# OpenReview forum: "Your Weak LLM is Secretly a Strong Teacher for Alignment"
_ICLR.cc/2025/Conference — ICLR 2025 Poster_

### Official Review · Reviewer_2XYr · 2024-10-18

**Soundness:** 3
**Presentation:** 3
**Contribution:** 2
**Rating:** 6
**Confidence:** 4

**Summary:**

This work proposes to investigate whether "weak" language models could serve as a reward model/preference generator for alignment. Experiments demonstrate that they can to varying extents and match some of the benefits of HF/AIF.

**Strengths:**

+ ailgnment with synthetic feedback/preference is an important research question
+ the experiments are comprehensive

**Weaknesses:**

- On the high-level framing, why is "weak LLM as feedback" positioned as the "middle ground" (line 42) between human feedback and AI/LM feedback? You could say that weak LLM feedback is still AI feedback, just with a smaller/weaker model, and not necessarily offers something strikingly new that make it "middle ground" in the "spectrum".

- By "weak LLM" the authors actually mean "small LLM": for example, "while a model like GPT-4 might have trillions of parameters,
a weak LLM might only have hundreds of millions or even fewer" (lines 44-45). The experiments focus on size too by employing OPT models, which are small compared to today's standards. However, "small LLM" for feedback is not necessarily "weak": if you take a small model and train it on large quantities of quality human preference data, it might become a moderate reward model. It would be great to disentangle "weak" and "small" in the narrative and also in the experiments.

- Section 2 is a bit trivial and takes up too much space. These math primers on DPO/PPO were not particularly helpful in later sections too. It would be great to assume readers of an "alignment" paper are already familiar with the basics, move them to the appendix, and use the space for analysis/results.

- I see that there is some investigation into the quality of preferences by the "weak" OPT model. Maybe something like RewardBench might offer a more rigorous quantitive evaluation.

- I would argue that things like self-rewarding language models [1] did things similar to "feedback from a weak model", albeit not as weak as OPT. Starting from a "weak-ish" models that are not good in both generating and evaluating, they show it is possible to jointly improve both. I wonder if the uniqueness of this paper's contribution might be weakened a bit by works like this.

- Ultimately, some might say that this work is a fine analysis paper on a niche technical design choice and does not offer something greater: taking RLAIF pipelines, changing the "AI" model part to a smaller model, and run some evaluation. Please highlight additional contribution if any.

[1] Yuan, Weizhe, et al. "Self-Rewarding Language Models." ICML 2024.

**Questions:**

please see above

---

> ### Author Response · Authors · 2024-11-18
> **Response to Reviewer 2XYr (Part I)**
>
> We thank the reviewer for the valuable and constructive feedback on our work. We are glad you recognize the importance of the problem and the comprehensiveness of our experiments. We address your comments and questions in detail below.
>
> > **W1: why frame weak LLM feedback as the "middle ground"?**
>
> Thank you for raising this question. We are happy to clarify further. We position weak LLM feedback as a "middle ground" because **our framework is inherently hybrid---the weak LLM is trained in a supervised manner utilizing some labeled human preferences, which then generates AI feedback on unlabeled dataset**. This positioning highlights a unique balance between scalability and cost efficiency. Human feedback faces scalability challenges, whereas feedback from large LLMs is highly scalable but computationally expensive. Weak LLM feedback provides a practical, scalable alternative that mitigates the trade-offs of both human and high-capacity LLM feedback, which represent two extremes on the feedback spectrum.
>
>
> > **W2: 'Weak' or 'small' LLM**
>
> That's an insightful point. We fully agree that "weak" and "small" imply different concepts and should be used with precision. In our study, the term "weak LLM" is intended to emphasize the limited capability of the preference model rather than size alone. Although our weak LLMs are indeed smaller in scale, they do not perform at the level of what might be considered a "moderate" reward model. We offer three pieces of evidence to clarify why our "weak LLM" is indeed weak:
>
> - First, our weak LLMs are not trained on extensive, high-quality human preference data. We use only a subset of the inherently noisy HH-RLHF dataset, which is far less comprehensive than large-scale datasets like UltraFeedback typically used to train high-performing reward models.
> - Second, as shown in Table 1, the weak LLM (OPT-125M) achieves only 60.6% accuracy (or consistency with human feedback label) on the test set of HH-RLHF, indicating its performance constraints.
> - Finally, our evaluation of the weak LLM’s preferences on RewardBench (details in response to W4) further confirms its constrained capabilities.
>
> For these reasons, we believe "weak LLM" accurately captures the intended model characteristics in our study. This also makes our finding novel and interesting, that using feedback from weak LLM can not only match but potentially exceed human feedback in alignment tasks.
>
> > **W3: Presentation of Section 2**
>
> Thank you for the suggestion on the presentation. Section 2 introduces essential mathematical notation and conventions specific to DPO/PPO, which we included to ensure clarity and consistency throughout the paper. While these details may enhance accessibility for a broader audience outside the alignment community, we agree that moving this content to the appendix would allow us to maintain a self-contained reference without diverting attention from our main analysis and results. We will incorporate this change.
>
> > **W4: Evaluate the weak LLM on the RewardBench**
>
> As suggested, we evaluate the weak LLM (OPT-125M) through the RewardBench, and report the results in the table below.
>
>
> | Chat | Chat Hard | Safety | Reasoning |Reasoning |
> | -------- | -------- | -------- | -------- |-------- |
> | 46.6     | 50.4    |41.3     |55.8     |48.5 |

---

> ### Author Response · Authors · 2024-11-18
> **Response to Reviewer 2XYr (Part II)**
>
> > **W5: Difference with self-rewarding language models**
>
> Our work aims to answer a fundamentally different research question from Yuan et al. [1] and bears several distinctions in problem formulation, methodology, and key findings.
>
> **_Difference in feedback source and mechanism_**. Yuan et al. start from the response of a relatively strong model, specifically Llama 2-70B, and utilize the same size model itself as both the generator and evaluator of responses. This model assesses its own outputs through an internal LLM-as-a-judge mechanism, effectively serving as a self-rewarding system. In contrast, our work employs a much weaker LLM to provide external rewards from the model as feedback to train a more powerful student model. Our central goal is to explore whether a model with weak capability can provide effective feedback for alignment, which is different from Yuan et al. In the setting of Yuan et al., the size of the student and teacher are consistent, while in our setting the student is weaker than the teacher.
>
> **_Difference in model training_**. The self-rewarding model in Yuan et al.'s work undergoes iterative training, refining its evaluation capabilities alongside its response generation. Our method, however, focuses on training the weak LLM specifically for feedback provision, without iterative self-improvement, and then assesses its effectiveness in guiding the student model's alignment.
>
> **_Difference in key finding_**. Yuan et al. demonstrate that a strong model can effectively self-align through internal feedback mechanisms. In contrast, our research demonstrates that feedback from a weak LLM can match or even surpass human feedback in certain alignment tasks. This finding suggests that smaller, less resource-intensive models can be effectively employed in alignment processes, offering a cost-efficient alternative to larger models.
>
>
>
> [1] Yuan, Weizhe, et al. "Self-Rewarding Language Models." ICML 2024.
>
>
>
> > **W6: Significance w.r.t. RLAIF**
>
> We appreciate your perspective. Our study offers several significant contributions that go beyond the scope of RLAIF.
>
> First, while RLAIF relies on pure AI feedback by prompting off-the-shelf models (e.g., GPT-4) to generate preference, our framework operates in a fundamentally different way. As discussed in response to W1, our framework is based on a hybrid approach, where the weak LLM is trained on a task-specific alignment dataset and produces preferences through a reward mechanism rather than simple prompting. Importantly, we find that substituting GPT-4 with a weak LLM in a prompt-based setup, as done in RLAIF, is not effective. Instead, our method is formulated as a semi-supervised alignment framework (Section 3.1), setting it apart from RLAIF’s problem formulation and design.
>
> Second, Section 4.5 of the RLAIF study suggests that alignment improves with larger AI labelers, which may (mis)lead the community to believe that smaller models are inadequate for providing effective feedback. Our findings challenge this notion by demonstrating that weak LLMs can serve as competent supervisors for stronger student models, offering reward signals that enable the student models to perform on par with, or even surpass, those trained with human or GPT-4 feedback. **To the best of our knowledge, this is the first time such a finding has been revealed**. _We believe such a finding is both novel and significant because it points the research community in an unexpected but promising direction. This opens new avenues for future research aimed at refining and expanding the use of weak signals in AI alignment strategies. This contributes to the broader discourse on developing efficient and effective methods for aligning AI systems with desired outcomes._
>
> Lastly, our paper offers more than evaluation---we conduct **in-depth understanding** on the effectiveness of weak LLM feedback for alignment in **Section 5**, shedding light on instances where it surpasses human feedback. Such a qualitative understanding is missing in RLAIF, which is our unique contribution. We believe this nuanced understanding is valuable to the research community, underscoring the potential of leveraging weak AI signals for scalable alignment solutions.
>
>
> | Method                   | Gold reward |
> |---------------------------------|-------|
> | Weak LLM (RLAIF style prompting)     | 3.87  |
> | GPT-4 (RLAIF style prompting)        | 4.52  |
> | Weak LLM (ours, reward mechanism)      | 4.63  |

---

> > ### Comment · Reviewer_2XYr · 2024-11-18
> >
> > I would like to thank the authors for their detailed response. As to the last point, yes there is much new "insights" and "understanding" of using weak tuned models for feedback, but the **methodology** itself might not be really strikingly new from self-rewarding, rlaif, etc. I still feel that this is "a fine analysis paper on a niche technical design choice and does not offer something greater", but raising the rating to 6.

---

> > > ### Author Response · Authors · 2024-11-18
> > > **Thanks for your follow up**
> > >
> > > Dear Reviewer 2XYr,
> > >
> > > Thank you for your thoughtful feedback and for raising the rating. We appreciate your recognition of the insights our work provides. We do position our paper primarily as an analysis-style contribution rather than a methodology-focused one. While we understand that opinions on what constitutes significant or novel contributions may vary, we firmly believe in the value of advancing understanding, which is equally important as introducing new methodologies.
> > >
> > > Thanks again!
> > > Authors.

---

### Official Review · Reviewer_ZoiK · 2024-10-25

**Soundness:** 4
**Presentation:** 4
**Contribution:** 3
**Rating:** 10
**Confidence:** 4

**Summary:**

This paper investigates the use of a weak supervised LLM to annotate preference data, as a middle ground between labor-intensive, costly human annotations and computationally expensive annotation based on high-capacity LLMs such as GPT-4. More specifically, the weak LLM (with approx. 100M parameters) is fine-tuned with DPO on a set of human labeled preference data and then its associated implicit reward model is applied to an unlabeled set to predict preferences. The resulting completed preference dataset is ultimately used to align a student LLM, whose performance enables measuring the quality of the annotations provided by the weak LLM.

The proposed weak LLM-based annotator is evaluated through multiple experiments where it is compared against both human annotators and stronger LLM annotators. It also includes experiments with weak LLMs and students models from different model families. Overall, the results show that using weak LLM-based annotations lead to similar or better performance compared to using human annotations or prompted high-capacity LLMs.

**Strengths:**

- The paper is clear, polished and reads very well.
- The experiments are comprehensive and convincing.
- The proposed idea of using a weak supervised LLM as annotator is likely to have a significant impact on future LLM alignment practices.

**Weaknesses:**

- The choice of using DPO fine-tuning to obtain a weak LLM-based annotator is not fully intuitive and lacks motivations, as only a reward model is needed for the preference annotation -- a model with generation capabilities is unnecessary for this (more details on this point in the "Questions" section).

**Questions:**

- In the second paragraph of the introduction (lines 40-50), expliciting the fact that what is studied is a weaker _supervised_ LLM might help the reader understand the middle-ground nature of this approach. Otherwise it may not be fully clear why weaker LLMs are positioned in between human annotators and high-capacity LLM annotators.
- In Figure 1, it is a bit unintuitive to show A < B in all the annotations, as it seems that answer A is a better (at least more helpful) answer than B in the given example. This example is mentioned later in Table 3, as a case where human annotation contradicts the gold reward model.  But using this (potentially incorrect) human annotation in Figure 1 when describing the approach might be confusing for the reader.
- The construction of $\mathcal{D}_{weak}$ in Section 3.1 is done by first learning a generation model through DPO and then use the implicit reward model induced by DPO to annotate the unlabeled dataset $\mathcal{D}_u$. It seems unnecessary to train a generation model as only the reward model is needed for the annotation. Given this, it is then unclear why DPO was used in this step instead of simply training a traditional reward model by following the standard RLHF protocol (but without the unneeded PPO stage). The argument given lines 139-140 that training with RLHF requires the use of multiple models is only valid for the PPO stage (when there are both the LLM policy and the reward model to handle), but not in the reward modeling stage which only consists of an LLM augmented with a regression head. It would have been interesting to study the impact of the DPO reward model vs the RLHF reward model.
- In Section 3.2, one of the metric introduced is the gold reward. However it is not entirely clear why a large auxiliary reward model can be considered as "gold". Does it just have to be a "recognized" reward model, or are there any requirements on the training data used for this reward model?
- For the GPT-4 win-rate metric, it does not seem necessary to ask for a numeric rating to be assigned to the two compared responses. Would it not be sufficient to just ask GPT-4 to identify the better response out of the two? This seems to be a more standard approach (see, e.g., Prometheus 2: https://arxiv.org/abs/2405.01535).
- What is the ratio $|\mathcal{D}_{l}|/|\mathcal{D}|$ used for the main experiments in Section 3.3? The ablations in Section 3.4 do study the results when this ratio is varied, but I could not find the default ratio that is used in other experiments.
- The x axis label of Figure 7 in Appendix C seems to be incorrect: it should be $\beta$ instead of "student model size".

---

> ### Author Response · Authors · 2024-11-18
> **Response to Reviewer ZoiK**
>
> We thank you for the positive and constructive feedback! We are deeply encouraged that you find our work clear, convincing, and likely to have a significant impact on future LLM alignment practices. Below we address your comments and questions in detail.
>
> > **W1/Q3: Comparison using the DPO instead of using the reward model**
>
> You raise an excellent question. Our choice to utilize DPO fine-tuning for weak LLM is grounded in two key considerations.
>
> First, by utilizing DPO, we maintain a **consistent and unified framework** throughout the entire training process, aligning both the reward model and the target student model under the same optimization paradigm. This consistency facilitates more straightforward integration in practice and is a common basis for the teacher-student framework (where both teacher and student models are optimized for a similar objective). Additionally, by fine-tuning the LLM with DPO, we effectively embed the reward model within the weak LLM itself. As you correctly noted, DPO inherently acts as an implicit reward model, since it's fundamentally formulated as a Bradley-Terry loss.
>
> That said, we recognize that when the weak LLM’s role is limited to producing reward signals, it is feasible to replace it with a traditional reward model. To explore this, we conducted additional experiments comparing DPO-trained weak LLMs with traditional reward models. The results indicate that both approaches yield comparable performance, with both surpassing the human labeler.
>
> | Feedback |Gold reward |
> | -------- | -------- |
> | Week LLM (w DPO)     | 4.84     |
> | Week LLM (w reward modeling)     | 4.86     |
> | Huamn labeler     | 4.63     |
>
>
> > **Q1: Clarification on introduction**
>
> Thank you for the great suggestion. We agree that clarifying the role of the “weaker _supervised_ LLM” will help readers better understand its middle-ground position. We will modify the introduction following your suggestion.
>
> > **Q2: The preferred answer in the Figure 1**
>
> Thank you for the careful read and for catching this issue! We understand how this could be confusing for readers, given the human annotation (following the HH-RLHF label) is potentially incorrect. We will replace it with another example where the preference is clear.
>
> > **Q4: Why call the "gold" reward model**
>
> The terminology of "gold reward" mostly follows the naming convention in literature. For example, previous work [1] referred to a strong reward model's reward as a "gold" reward. The model is considered "gold" due to its comprehensive training on extensive, high-quality preference datasets, enabling it to serve as a proxy for assessing response quality. Considering that the reward for answer quality in the real world is difficult to simulate, we choose to use the SOTA reward model in RewardBench to simulate the gold reward for evaluation.
>
> [1] Scaling laws for reward model overoptimization ICML 2023
>
> > **Q5: GPT-4 evaluation with the prompt to directly identify the better response out of the two**
>
> That's an interesting suggestion. Following your suggestion, we perform the GPT-4 evaluation using the prompt from Prometheus 2 [1], directly asking GPT-4 to identify the better response out of the two. We report the win-rate in the table below on different models. Similar to our main experiments, the weak LLM feedback is based on OPT-125M. The win rate for a model trained with weak LLM feedback approaches 50% for all cases, indicating that its performance competitively matches that of using human feedback.
>
>
> | Student model | OPT-1.3B | OPT-2.7B |Mistral-7B |
> | -------- | -------- | -------- | -------- |
> | Win-rate     | 52    |49     |51    |
>
> [1] Kim, Seungone, et al. "Prometheus 2: An open-source language model specialized in evaluating other language models." arXiv preprint arXiv:2405.01535 (2024).
>
> > **Q6: Default ratio used in experiments**
>
> The default ratio used in experiments is 1/2. We have added this to our manuscript - thanks for catching this issue!
>
>
> > **Q7: The wrong x axis label of Figure 7**
>
> Thanks again for your careful read! We have fixed the x axis label in Figure 7 as $\beta$.

---

> > ### Comment · Reviewer_ZoiK · 2024-11-21
> > **Response to authors**
> >
> > Many thanks for the detailed and very clear response. I greatly appreciated the additional experiments conducted by the authors, namely using traditional reward modeling instead of DPO to obtain the weak LLM annotator, and the evaluation based on the prompt from Prometheus-2. The results confirm the stability of the approach in different settings. I can also understand the motivation to keep a consistent framework (DPO) to train both the student and teacher model.
> >
> > I have read as well the other reviews and the thorough responses provided by the authors. All in all, I am further convinced me of the high value of this work and its potential for impact on LLM alignment research. For this reason, I am upgrading my score to 10.

---

> > > ### Author Response · Authors · 2024-11-21
> > >
> > > Dear reviewer ZoiK,
> > >
> > > Thank you for your thoughtful feedback and for upgrading the score to 10—they truly mean a lot to us and our hard work. We deeply appreciate your recognition of the stability and potential impact of our approach and your support for its value to the community!
> > >
> > > Sincerely,
> > >
> > > Authors

---

### Official Review · Reviewer_ntey · 2024-10-30

**Soundness:** 3
**Presentation:** 4
**Contribution:** 2
**Rating:** 6
**Confidence:** 4

**Summary:**

The paper explores the potential of using weak large language models (LLMs) for AI alignment. It proposes a new framework that leverages weak LLMs to provide feedback, thereby reducing reliance on costly human labor and computational resources. Experimental results show that even weak LLMs with as few as 125 million parameters can offer feedback comparable to human feedback. The paper also analyzes the quality of the feedback from weak LLMs and finds that it can surpass human judgment in some cases, although there is uncertainty in fine differentiation. The study offers a new perspective for AI alignment and demonstrates that weak LLM feedback is a promising, scalable and sustainable solution.

**Strengths:**

1.The paper introduces a framework that leverages weak LLMs for alignment, which offers a promising alternative to existing methods that rely of either extensive human labor or expensive computational resources.

2.The paper conducts comprehensive evaluation of weak LLM feedback across various model scales and families. The result consistently demonstrate the effectiveness of this approach, highlighting its potential for practical application.

3.The paper is well-organized, with clear sections and logical flow. The authors effectively communicate the research objectives, methodology, and findings.

4.The research has significant practical implications for the development and deployment of aligned LLMs. Utilizing weak LLM feedback can reduce costs and improve efficiency, making alignment more scalable and accessible.

**Weaknesses:**

1.In this paper， the authors propose a framework that utilizes weak LLMs for alignment, striving to strike a balance between RLHF and RLAIF, it appears that the weak-to-strong approach and methodology in the paper are still quite similar to Burns et al. (2024). Burns et al. (2024)’s study focus on reward modeling and binary classification, where the outputs in this paper are either scalar or categorical labels. Your study is closely tied to alignment and involves a more complex output space. However, it seems that the two are consistent in their methodological approach. Therefore, I think that there are still some limitations in terms of novelty. I hope you can explain more about the difference between the two works.

2.The paper points out that even advanced LLMs like GPT-4 may show feedback inconsistency when the quality difference between two responses is subtle. In the case where it is uncertain whether gold RM is stronger than GPT-4, when using gold RM for evaluation, is it also possible to make incorrect judgments about two responses with similar quality, leading to bias in the evaluation？Additionally，When using GPT-4 for evaluation, although the prompt already mentions and ensures that the order of the responses should not affect its judgment, in actual experiments, it is difficult to avoid the model's preference caused by the order. A fairer approach would be to swap the positions of the two responses and conduct the experiment again. I believe that conducting the experiment twice with different positions would result in a more accurate evaluation.

3.While the paper demonstrates the effectiveness of weak LLM feedback across alignment task, it would be beneficial to investigate its applicability to more complex tasks.

**Questions:**

1.See weakness 2.

2.See weakness 3. Can this weak-to-strong approach be evaluated for more complex tasks? Like a math task or a code task?

3.In Contribution 2, the authors mentioned that using weak LLMs to provide preference labels for alignment can match or even surpass the performance of using full human feedback with the same training data scale. Why is it that training on data labeled by weak LLMs on unlabeled data can outperform the training set labeled by humans? Is it because the human-labeled training set contains some incorrect labels and its quality still needs improvement?

---

> ### Author Response · Authors · 2024-11-18
> **Response to Reviewer ntey (Part I)**
>
> We thank the reviewer for the thorough and constructive feedback. We are encouraged that you recognize the significance of our work. Below we address your comments and questions in detail.
>
> > **W1: Difference w.r.t. Burns et al. (2024)**
>
> We discuss the differences w.r.t. Burns et al. (2024) in **Section 5** (**L461-L476**). To recap, there are several major distinctions to note.
>
> - [**Problem formulation is different**] First, as you noted, while Burns et al. focused on simpler tasks like reward modeling and binary classification with scalar or categorical outputs, we tackle the more complex task of language generation and alignment, which involves a significantly more sophisticated output space. This shift in focus introduces a novel exploration of weak LLM feedback for alignment---a direction Burns et al. did not address.
> - [**Evaluation framework is different**] Additionally, while Burns et al. evaluated their models with accuracy metrics suitable for classification tasks, we use a different evaluation framework tailored to the generative nature of the alignment task. This involves using gold reward models and GPT-4 evaluations and performing an in-depth qualitative study to understand the human vs. LLM feedback (Section 4). All of our experimental results and analysis sections are novel.
> - [**Conclusion is different/opposite**] Lastly, we draw an entirely opposite conclusion from Burns et al. In particular, they reported models trained with weak LLM feedback are noticeably worse than those trained with human feedback, especially for reward modeling, our findings challenge these conclusions. Considering that human preferences are noisy and unreliable, accuracy on the preference test set for alignment is a questionable metric. Our works show contrary conclusions when directly measuring the generation performance of the aligned model, and demonstrate **for the first time** that _using feedback from weak LLM can not only match but potentially exceed human feedback in alignment tasks_. Contrary to the pessimistic view presented by Burns et al., our results advocate for the promising potential of leveraging weak AI signals for alignment. This opens new avenues for future research aimed at refining and expanding the use of weak signals in AI alignment strategies.
>
>
> > **W2: Potential bias in evaluation using gold RM and GPT-4**
>
> This is an excellent question, and we address it in three parts.
>
> First, it's important to acknowledge that using model-based approaches for evaluating open-ended generations remains a significant research challenge. Although GPT-4 and gold reward models are not flawless (as shown in our study), they are widely used in the literature due to their scalability and strong correlation with human preferences.
>
>
> To further enhance the reliability of evaluation, we incorporate multiple top-performing reward models from RewardBench [1]. This consists of **several independently trained reward models, helping to mitigate biases and errors from any single model**. We present a table where each row corresponds to a different reward model highly ranked in RewardBench. This approach demonstrates that models trained with weak LLM feedback either match or surpass those trained with human feedback across various evaluation metrics, providing a more balanced and comprehensive assessment.
>
> | Reward model | Weak LLM feedback | Human feedback|
> | -------- | -------- | -------- |
> |Skywork/Skywork-Reward-Llama-3.1-8B-v0.2| -1.25 | -1.30 |
> |Ray2333/GRM-Llama3-8B-rewardmodel-ft | 0.25 | 0.16 |
> |RLHFlow/ArmoRM-Llama3-8B-v0.1|1.27|1.42|
> |Ray2333/GRM-gemma2-2B-rewardmodel-ft|1.95|2.02|
> |NCSOFT/Llama-3-OffsetBias-RM-8B| 8.91 | 8.61     |
>
>
>
> Lastly, **we complemented our model-based evaluations with human evaluations** to assess the quality of the responses generated by models trained with weak LLM feedback versus those trained with human feedback. The average win-rate across 100 pairs of responses is 52%, closely aligning with our main findings that weak LLM feedback is at least as effective as human feedback.
>
> [1] Lambert, Nathan, et al. "RewardBench: Evaluating Reward Models for Language Modeling, March 2024." URL http://arxiv.org/abs/2403.13787.

---

> ### Author Response · Authors · 2024-11-18
> **Response to Reviewer ntey (Part II)**
>
> > **W2/Q1: Swap the positions of the two responses and conduct the experiment.**
>
> We completely agree with your concern! In fact, our implementation and evaluation already incorporate the suggested approach: we randomly swap the positions of the two responses with a probability of 0.5 before inputting them into GPT-4. This means that outputs from the model trained with weak LLM feedback and the one trained with human feedback **are equally likely to appear in either position**. We will make sure to clarify this in our manuscript - thanks again for catching this issue.
>
> > **Q2/W3: Evaluation on more complex task**
>
> As suggested, we additionally evaluate the effect of our framework by testing it on the task of generating code from natural language, using Spider [1], an open-source dataset for SQL code generation. We collected model error outputs paired with ground truth code, which are used as code preference dataset. Then we use the method proposed in Section 3 to collect code preference annotated with weak LLM (OPT-125M). This dataset was compared against feedback generated by GPT-4 using the RLAIF framework.
>
> The performance of the OPT-1.3B student model under different feedback sources is summarized in table below. The results show that feedback from the trained weak LLM significantly enhances the student model's code generation ability compared to directly prompting GPT-4 for code preferences. This demonstrates the ability of weak LLM feedback to generalize effectively to more complex tasks.
>
>
> | Feedback               | Execution accuracy |
> |------------------------|--------------------|
> | Weak LLM feedback      | **45.2**%             |
> | GPT-4 feedback (RLAIF) | 39.9%             |
>
>
>
>
> [1] Yu, Tao, et al. "Spider: A large-scale human-labeled dataset for complex and cross-domain semantic parsing and text-to-SQL task." arXiv preprint arXiv:1809.08887 (2018).
> > **Q3: Why is it that training on data labeled by weak LLMs on unlabeled data can outperform the training set labeled by humans?**
>
> Thank you for this insightful question. We provide an in-depth analysis of weak LLM feedback quality in **Section 4**. Specifically, in **L388-L414**, we examine the quality differences between weak LLM and human feedback. In cases where weak LLM feedback diverges from human feedback, we find that in 44.3% of these instances, the response chosen by the weak LLM has a higher gold reward score. This result suggests that weak LLM preferences are superior to human feedback in nearly half of these cases. As you pointed out, the comparable performance between weak LLM-labeled and human-labeled data can be attributed to the fact that human feedback is inherently imperfect and often contains noise. We include additional qualitative examples in **Appendix B** to further demonstrate the noise in human feedback.

---

> > ### Author Response · Authors · 2024-11-22
> >
> > Dear reviewer ntey,
> >
> > We wanted to touch base with you as the deadline for the author-reviewer discussion phase is approaching on November 26. We trust you've had the opportunity to review our rebuttal, and we would be more than happy to address any further concerns you have.
> >
> > Thank you once again for your time and dedication to this review process. We look forward to your response and to furthering the dialogue on our manuscript.
> >
> > Best,
> >
> > Authors

---

> > > ### Comment · Reviewer_ntey · 2024-11-23
> > >
> > > Thank you for your detailed feedback. The experiment is comprehensive, and the results are very detailed. I think your work has great potential and certain enlightenments for LLM alignment. I will raise the score.

---

> > > > ### Author Response · Authors · 2024-11-23
> > > > **Thanks for your follow up**
> > > >
> > > > Dear Reviewer ntey,
> > > >
> > > > Thank you for your thoughtful feedback and for raising the rating. We appreciate your recognition of the insights our work provides!
> > > >
> > > > Thanks again.
> > > > Authors

---

### Official Review · Reviewer_oS4F · 2024-11-03

**Soundness:** 4
**Presentation:** 3
**Contribution:** 3
**Rating:** 6
**Confidence:** 3

**Summary:**

This paper introduces an alignment approach using weak large language models (LLMs) that balances cost and efficiency. The authors find that weak LLMs can generate feedback comparable to or even better than human feedback and stronger LLMs and propose the method to employ a semi-supervised framework where a weak LLM, trained on a small labeled dataset, provides preference feedback for a larger unlabeled dataset, reducing reliance on human annotation.

**Strengths:**

1. This paper proposes a cost-effective alignment framework that reduces computational costs and reliance on extensive human feedback,  using weak LLMs that have significantly fewer parameters than high-capacity models.

2. The study demonstrates that weak LLMs can perform as well as, and in some cases better than, human annotators in generating preference feedback for alignment through extensive experiments. For example, by demonstrating similar alignment success across different model families and sizes, the study provides a strong case for the generalizability of the weak LLM approach, suggesting that high alignment quality can be achieved without the latest or most powerful models.

**Weaknesses:**

1. The paper lacks a thorough analysis of feedback consistency over repeated evaluations. For example, when weak LLM and human preferences conflict, the study does not adequately explore the stability of weak LLM feedback. A quantitative analysis of consistency across multiple runs or various weak models would be helpful to assess whether the weak LLM’s feedback is repeatable and reliable.

2. The evaluation metrics are only based on 2 models, the gold reward model and GPT-4. If the models are biased, the evaluation given by them would be unreliable and it could lead to skewed assessments that do not accurately reflect the quality of weak LLM alignment.

**Questions:**

1. Is the consistency of weak LLM feedback assessed across multiple prompts or model runs?

2. How do the authors ensure that the gold reward model and GPT-4 are reliable in evaluation? Are other potential evaluation metrics considered?

3. The paper listed many LLM alignment methods in the related work. Did the authors compare their method with these alignment techniques?

---

> ### Author Response · Authors · 2024-11-18
> **Response to Reviewer oS4F**
>
> We thank the reviewer for the insightful feedback and positive evaluation of our work. We appreciate the acknowledgment of our approach's potential to reduce both computational costs and reliance on extensive human feedback. We address your comments and questions in detail below.
>
> > **W1/Q1: Consistency of weak LLM feedback**
>
> As suggested, we perform additional investigation by evaluating the consistency of weak LLM feedback across multiple runs. Specifically, we **trained 5 instances of the weak LLM (OPT-125M) with different seeds**, resulting in a set of varied weak LLMs. Across different samples, the variance in feedback preference was measured at 0.04, which suggests a relatively low variance and a consistent alignment in feedback for the weak LLM trained with different seeds. This result demonstrates that the weak LLM can reliably provide consistent preferences across different runs.
>
> > **W2/Q2: How do the authors ensure that the gold reward model and GPT-4 are reliable in evaluation? Are other potential evaluation metrics considered?**
>
> This is an excellent question, and we address it in three parts.
>
> First, it's important to acknowledge that using model-based approaches for evaluating open-ended generations remains a significant research challenge. Although GPT-4 and gold reward models are not flawless, they are widely used in the literature due to their scalability and strong correlation with human preferences [1].
>
>
> To further enhance the reliability of evaluation, we incorporate multiple top-performing reward models from RewardBench [2]. This consists of **several independently trained reward models, helping to mitigate biases and errors from any single model**. We present a table where each row corresponds to a different reward model highly ranked in RewardBench. This approach demonstrates that models trained with weak LLM feedback either match or surpass those trained with human feedback across various evaluation metrics, providing a more balanced and comprehensive assessment.
>
> | Reward model | Weak LLM feedback | Human feedback|
> | -------- | -------- | -------- |
> |Skywork/Skywork-Reward-Llama-3.1-8B-v0.2| -1.25 | -1.30 |
> |Ray2333/GRM-Llama3-8B-rewardmodel-ft | 0.25 | 0.16 |
> |RLHFlow/ArmoRM-Llama3-8B-v0.1|1.27|1.42|
> |Ray2333/GRM-gemma2-2B-rewardmodel-ft|1.95|2.02|
> |NCSOFT/Llama-3-OffsetBias-RM-8B| 8.91 | 8.61     |
>
>
>
> Lastly, **we complemented our model-based evaluations with human evaluations** to assess the quality of the responses generated by models trained with weak LLM feedback versus those trained with human feedback. The average win-rate across 100 pairs of responses is 52%, closely aligning with our main findings that weak LLM feedback is at least as effective as human feedback.
>
>
> [1] Liu, Yang et al. "G-eval: Nlg evaluation using GPT-4 with better human alignment." arXiv preprint arXiv:2303.16634 (2023).
>
> [2] Lambert, Nathan, et al. "RewardBench: Evaluating Reward Models for Language Modeling, March 2024." URL http://arxiv.org/abs/2403.13787.
>
>
> >**Q3: The paper listed many LLM alignment methods in the related work. Did the authors compare their method with these alignment techniques?**
>
> We included an extensive list of alignment approaches in our related work section to provide readers with a comprehensive overview of this actively developing field. _While most alignment methods discussed in the related work focus primarily on training strategies, our approach distinctly emphasizes the feedback sources, which is orthogonal to alignment techniques_. Our experiments are designed to highlight the role of feedback sources rather than the alignment approaches themselves. Specifically, we benchmarked the performance of our method against both human feedback-based methods and other AI-driven feedback methods (such as RLAIF) discussed in the related work.
>
>
> With that being said, exploring the use of weak LLM feedback in conjunction with more advanced and sophisticated alignment approaches presents a compelling avenue for future work. We believe integrating our approach with these more complex alignment strategies could yield even more promising performance, enhancing both efficiency and effectiveness in model alignment.

---

> > ### Comment · Reviewer_oS4F · 2024-11-22
> > **Response to Authors**
> >
> > Thank the authors for the detailed response! I will increase the soundness to 4.

---

> > > ### Author Response · Authors · 2024-11-23
> > > **Thanks for the follow up**
> > >
> > > Dear Reviewer oS4F,
> > >
> > > Thank you for your thoughtful feedback and for raising the soundness. We appreciate your recognition of the insights our work provides!
> > >
> > > Thanks again.
> > > Authors

---

### Official Review · Reviewer_1dzH · 2024-11-03

**Soundness:** 3
**Presentation:** 3
**Contribution:** 2
**Rating:** 6
**Confidence:** 4

**Summary:**

This paper concentrates on the critical research topic of large language model (LLM) alignment, especially related to super-alignment. Considering existing mainstream alignment approaches rely on paired human preference data, which highly relies on human or powerful LLMs for annotation, using a weak LLM to provide alignment feedback becomes a promising alternative. This paper aims to bridge that gap of lacking a systematic evaluation and understanding of weak LLM’s capability of generating feedback of alignment. The authors first formalize a workflow to conduct semi-supervised alignment training of a model and conduct evaluation. Based on this uniformed experimental setting, the paper assesses the effectiveness of models of various scales in providing alignment feedback and finds that weaker LLMs can indeed play an effective role in alignment tasks. Additionally, an in-depth analysis is conducted on the quality of feedback provided by weaker LLMs.

**Strengths:**

- The research topic in this paper is critical and popular. Given the high costs associated with human and large model annotations, exploring the potential of weaker LLMs in providing alignment feedback is a timely and necessary endeavor.
- This paper is well-written and easy-to-follow, with an introduction to associated alignment background techniques.
- A dedicated discussion section highlights the differences between this work and the most relevant prior studies.

**Weaknesses:**

- More comprehensive experiments are required to validate the reliability of the primary conclusions in this paper, including but not limited to the following aspects.
    + The most relevant baselines with the idea of RLAIF. Beyond that the original RLAIF methods rely on large LLMs to provide feedback, it is crucial to delineate differences between the proposed training pipeline and RLAIF. Additionally, a comparison with RLAIF employing weaker LLMs as the reward model would strengthen the findings.
    + Human annotation data of different quality levels. Previous studies [1,2] have indicated that the HH-RLHF dataset may contain noise in its human preference labels. Comparing with higher-quality human data, such as a revised version of HH-RLHF, would enhance the robustness of the results.
    + LLMs with larger sizes.
    + GPT-4 with more advanced prompts or instructions to provide alignment feedback.

[1] Wang, Binghai, et al. "Secrets of rlhf in large language models part ii: Reward modeling." arXiv preprint arXiv:2401.06080 (2024).

[2] Liu, Yan, et al. "Elephant in the Room: Unveiling the Impact of Reward Model Quality in Alignment." arXiv preprint arXiv:2409.19024 (2024).

- The issue of overfitting to a specific dataset in smaller models is not discussed. Since small LLMs are prone to overfitting to specific datasets but LLMs alignment target diverse domains, you should consider the generalization of the weaker LLM to provide alignment across domains. If it shows very weak generalization, it may need annotation data in each domain, which also requires a large human cost.
- More diverse and more reliable evaluation metrics should be considered.
- The actionable insights derived from the in-depth analysis are sparse. The main finding is that issues persist in the original annotated data, and both weaker LLMs and large models like GPT struggle to handle noisy data with high confidence.

**Questions:**

- Why the weaker LLM to provide alignment feedback is referred to as a task-specific model? It is trained with the HH-RLHF dataset, which is a general domain dataset for LLM alignment.
- What does the “pre-trained strong model” mentioned in Line 371 specifically refer to? Is is the LLM that only undergone pre-training without fine-tuning or alignment, or is it the trained with positive examples?
- I have concern about the result that GPT-4 underperforms compared to smaller models. Could you please provide more reasonable explanations?

---

> ### Author Response · Authors · 2024-11-18
> **Response to Reviewer 1dzH (Part I)**
>
> We thank the reviewer for the thorough and insightful feedback. We appreciate that you recognize the importance of the research problem, our clear presentation, and our positioning with respect to prior studies. We address your additional comments below.
>
> > **W1.1: Differences w.r.t. RLAIF, comparison with RLAIF employing weaker LLMs as the reward model.**
>
> We fully agree with you on this point. We already dedicated such a discussion and a systematic comparison in our manuscript **L267-L297** and **Figure 2(b)**. For your convenience, we elaborate on the differences below and provide additional experiments.
>
> First, while RLAIF relies on pure AI feedback by prompting off-the-shelf models (e.g., GPT-4) to generate preference, our framework operates in a fundamentally different way. In particular, our framework is based on a hybrid approach, where the weak LLM is trained on a human-labeled alignment dataset and produces AI feedback through inherent reward mechanisms rather than simple prompting. Our method is formulated as a semi-supervised alignment framework (Section 3.1), setting it apart from RLAIF’s problem formulation and design. Importantly, we find that substituting GPT-4 with a weak LLM (OPT-125M) in a prompt-based setup, as done in RLAIF, is not effective.
>
>
>
> Moreover, Section 4.5 of the RLAIF study suggests that alignment improves with larger AI labelers, which may (mis)lead the community to believe that smaller models are inadequate for providing effective feedback. Our findings challenge this notion by demonstrating that weak LLMs can serve as competent supervisors for stronger student models, offering reward signals that enable the student models to perform on par with, or even surpass, those trained with human or GPT-4 feedback. **To the best of our knowledge, this is the first time such a finding has been revealed to the research community**. _We believe such a finding is both novel and significant because it points the research community in an unexpected but promising direction. This opens new avenues for future research aimed at refining and expanding the use of weak signals in AI alignment strategies. This contributes to the broader discourse on developing efficient and effective methods for aligning AI systems with desired outcomes._
>
> | Method                   | Average Gold reward |
> |---------------------------------|-------|
> | Weak LLM (RLAIF style prompting)     | 3.87  |
> | GPT-4 (RLAIF style prompting)        | 4.52  |
> | Weak LLM (ours, reward mechanism)      | 4.63  |
>
>
> > **W1.2 Higher-quality preference dataset**
>
> As suggested, we conduct additional experiments by employing the revised version of HH-RLHF [1] for evaluation. We select the high-quality samples that are in the top 70 percentile in terms of the mean preference difference. We report the gold reward of the target model (OPT-1.3B) trained with the weak LLM feedback (OPT-125M) and human feedback. **Our conclusion holds consistently on such higher-quality human preference dataset**, where using a weak LLM to provide a preference label for alignment can match or even exceed the performance of using full human feedback.
>
>
> | Feedback              | Gold reward |
> |---------------------------|-------|
> | Weak LLM feedback     | 5.23  |
> | Human feedback        | 5.19  |
>
>
> [1] Wang, Binghai, et al. "Secrets of rlhf in large language models part ii: Reward modeling." arXiv preprint arXiv:2401.06080 (2024).
>
> > **W1.3: LLMs with larger sizes**
>
> We agree on the importance of validating our findings across different model sizes. We already provide the results with larger-size target models in **Figure 2(a)** and **Figure 3(a)**, including OPT-1.3B, OPT-2.7B, OPT-6.7B, OPT-13B, Mistral-7B, Llama-2-7B, and Gemma-7B. Additionally, we provide results in **Figure 2(b)** using larger-size models as the supervisor, including OPT-1.3B, Llama-3-8B, and GPT-4. Due to computing resource constraints, the largest model we were able to train is of size 13B.
>
> During rebuttal, we conducted more experiments by taking the Mistral-7B as the target model, and comparing its performance trained with different sources of feedback: OPT-125M, OPT-1.3B, OPT-2.7B, and human feedback. **Our conclusion holds consistently** that _using a weak LLM, with a size as small as 125M, to provide preference feedback for alignment can match or even exceed the performance of using pure human feedback_.
>
> | Feedback              | Gold reward |
> |---------------------------|-------|
> | Weak LLM (OPT-125M)     | 7.90  |
> | Weak LLM (OPT-1.3B)     | 7.82  |
> | Weak LLM (OPT-2.7B)     | 7.96  |
> | Human Labeler        | 7.19  |

---

> ### Author Response · Authors · 2024-11-18
> **Response to Reviewer 1dzH (Part II)**
>
> > **W1.4: GPT-4 with more advanced prompts to provide alignment feedback**.
>
> This is a great idea! As suggested, we use the more advanced prompt from Prometheus 2 [1] to provide alignment feedback. We report the comparison in the table below, where we measure the performance of the target model (OPT-1.3B) under different sources of feedback. **The adoption of a more advanced prompt does improve the average gold reward from 4.52 (RLAIF) to 4.76, though weak LLM feedback remains competitive**.
>
> | Method              | Gold reward |
> |---------------------------|-------|
> | GPT-4 (same prompt as RLAIF)    | 4.52  |
> | GPT-4 (advanced prompt)    | 4.76  |
> | Weak LLM (OPT-125M)     | **4.84**  |
> | Human feedback        | 4.63  |
>
>
> [1] Kim, Seungone, et al. "Prometheus 2: An open-source language model specialized in evaluating other language models." arXiv preprint arXiv:2405.01535 (2024).
>
> > **Q1/W2: Why the weaker LLM to provide alignment feedback is referred to as a task-specific model? What about the generalization aspect?**
>
> The weaker LLM is referred to as a task-specific model because preference datasets are designed with distinct criteria tailored to specific objectives. For example, HH-RLHF defines preferred responses as those that are both helpful and harmless, aligning the model with these particular goals. Conversely, datasets like TL;DR prioritize concise and accurate summarization, reflecting different definitions of "preference." As each dataset encodes its own standards, models trained on these datasets inherently align with the specific tasks they target. _This task-specific alignment explains why a model trained on HH-RLHF should not be expected to distinguish preferences for TL;DR, as the two tasks are guided by fundamentally different criteria_.
>
>
> Furthermore, we clarify that our framework is not aimed at building a general-purpose weak LLM. Instead, it is designed for scenarios where a small amount of human-labeled data is available, offering a significantly more cost-effective alternative to full human supervision. For any given task, a weak LLM trained on this subset of human preference data can provide feedback to achieve alignment performance comparable to full human supervision, yielding substantial savings in annotation costs. While cross-task evaluation is an interesting avenue for general-purpose reward models, it is outside the scope of our approach.
>
>
> Lastly, if the objective is to create a general-purpose labeler capable of generalizing across diverse domains and tasks, a model like GPT-4 would be more suitable. However, there is a tradeoff: general-purpose models such as GPT-4 excel in cross-task generalization but are computationally expensive and can be suboptimal for task-specific alignment. In contrast, our framework gives practitioners an additional choice. When working with task-specific datasets and limited human feedback, our method provides an efficient, scalable alternative, allowing practitioners to decide whether to leverage a general-purpose model like GPT-4 or rely on our framework tailored for resource-constrained settings.

---

> ### Author Response · Authors · 2024-11-18
> **Response to Reviewer 1dzH (Part III)**
>
> > **W3: More diverse and more reliable evaluation metrics should be considered.**
>
> This is an excellent point, and we address it in three parts.
>
> First, it's important to acknowledge that using model-based approaches for evaluating open-ended generations remains a significant research challenge. Although GPT-4 and gold reward models are not flawless (as shown in our study), they are widely used in the literature due to their scalability and strong correlation with human preferences.
>
> To further enhance the reliability of evaluation, we incorporate multiple top-performing reward models from RewardBench [1]. This consists of **several independently trained reward models, helping to mitigate biases and errors from any single model**. We present a table where each row corresponds to a different reward model highly ranked in RewardBench. This approach demonstrates that models trained with weak LLM feedback either match or surpass those trained with human feedback across various evaluation metrics, providing a more balanced and comprehensive assessment.
> | Reward model | Weak LLM feedback | Human feedback|
> | -------- | -------- | -------- |
> |Skywork/Skywork-Reward-Llama-3.1-8B-v0.2| -1.25 | -1.30 |
> |Ray2333/GRM-Llama3-8B-rewardmodel-ft | 0.25 | 0.16 |
> |RLHFlow/ArmoRM-Llama3-8B-v0.1|1.27|1.42|
> |Ray2333/GRM-gemma2-2B-rewardmodel-ft|1.95|2.02|
> |NCSOFT/Llama-3-OffsetBias-RM-8B| 8.91 | 8.61     |
>
> Lastly, **we complemented our model-based evaluations with human evaluations** to assess the quality of the responses generated by models trained with weak LLM feedback versus those trained with human feedback. The average win-rate across 100 pairs of responses is 52%, closely aligning with our main findings that weak LLM feedback is at least as effective as human feedback.
>
> [1] Lambert, Nathan, et al. "Rewardbench: Evaluating reward models for language modeling." .
>
> > **W4: Actionable insights from in-depth analysis.**
>
> Our in-depth analysis in Section 4 provides several important insights that advance our understanding of alignment and inform future research directions.
>
> First, we delve into the quality of weak LLM versus human feedback, offering direct explanations for **why weak LLM feedback performs as effectively as human feedback for alignment, which underpins our main findings**. Notably, our analysis reveals that in nearly half the cases where weak LLM feedback diverges from human feedback, the weak LLM's choices surpass human annotations in quality, **highlighting its potential to complement and even enhance human feedback**.
>
> **The analysis also sheds light on the importance of addressing noise in the human-annotated data**. By identifying weaknesses in existing datasets, we advocate for improved data curation and preprocessing strategies, which can in turn enhance the effectiveness of training weak LLMs to provide feedback. This aligns with our demonstration in response to W1.2.
>
>
> Moreover, with more qualitative analysis, we reveal weak LLM feedback and human feedback diverge significantly in cases with subtle quality distinctions. Advanced models like GPT-4 show low preference consistency in these scenarios, reflecting the inherent challenge of distinguishing between closely matched responses. **This points to the need for improved mechanisms to handle ambiguity in feedback collection, offering actionable directions for future research to tackle these nuanced cases.**
>
> Due to space constraints, we include additional qualitative analyses in **Appendix C**, such as (1) understanding systematic vs. random noise in weak LLM feedback, (2) correlations between generation quality in models trained with weak LLM versus human feedback, and (3) the impact of the KL coefficient on alignment performance when leveraging weak LLM feedback. We believe these insights provide valuable contributions to the community.
>
> We will revise the manuscript to ensure these actionable insights are more prominently discussed and effectively linked to their practical implications for advancing alignment research.
>
> > **Q2: The meaning of “pre-trained strong model"**
>
> The pre-trained strong model in line 371 refers to the model without any fine-tuning or alignment.
>
> > **Q3: Why GPT-4 underperforms compared to smaller models.**
>
> That's an insightful question! While directly prompting GPT-4 to generate preferences demonstrates significant potential, it is still prone to biases, which can negatively impact preference quality. Additionally, GPT-4 is not explicitly optimized to distinguish preferences for specific tasks, unlike our weak LLM, which is trained on task-specific data to align with particular objectives. Our experimental results show that task-specific LLMs provide more reliable feedback, leading to improved alignment performance compared to relying solely on general-purpose models like GPT-4. Our qualitative analysis in Section 5 (**Lines 415-448**) further highlights the unreliability of GPT-4’s feedback in certain contexts.

---

> ### Comment · Reviewer_1dzH · 2024-11-19
> **Response to Author Rebuttal**
>
> Thank you very much for the detailed responses and the extensive additional experiments, which have addressed some of my concerns. However, I still have two major concerns:
> 1. In your response to Q1/W2, you explained that, compared to GPT-4, you approach trains a weaker LLM on task-specific data to provide better signal for aligning with particular objectives. Thus, your weaker LLMs yield better performance on specific domains while GPT-4 would be more suitable in general-purpose settings. I think this explanation reasonable, however, this scope and limitation were not clearly stated in your paper, which risks misleading readers. Furthermore, if your method is only effective in specific scenarios, its contribution and significance may appear less compelling compared to general-purpose approaches like RLAIF.
>
>
> 2. While I agree that using a weaker LLM has potential efficiency advantages, I remain skeptical of its claimed superiority in effectiveness. The supplementary experiments provided additional evidence:
> - GPT-4, with advanced prompts, shows promising results that are comparable to those of the weaker LLM.
> - When evaluated on higher-quality human-labeled data, the observed improvement is much less pronounced.
> - Not all evaluation metrics consistently support the superior performance of the weaker LLM.
> - The effectiveness of weaker LLM also increases as the scale gets larger, from OPT(125M) to OPT-1.3B and Llama-3-8B.
>
> Moreover, the paper does not include comparisons where a large-scale LLM is trained in your framework as the supervisor (W1.3).
> Based on these points, I find the claimed effectiveness advantage of the weaker LLM to be unconvincing.

---

> ### Author Response · Authors · 2024-11-20
> **Follow up to Reviewer 1dzH**
>
> We thank the reviewer for taking the time to read our response and actively engaging in this dialogue. We are happy to clarify the concerns further.
>
> **Our paper is not about establishing performance superiority** but rather about **demonstrating the promise** of weak LLMs for alignment relative to human feedback. The central focus of our study is the contrast between weak LLM vs. human feedback, as reflected in our primary claim:
>
> > "_**The alignment performance using weak LLM feedback closely matches or even surpasses that of using human feedback**._"
>
> We carefully chose wording such as "closely match" to avoid implying outright superiority. Below, we address your concerns in more detail.
>
> --------------
> > General-purpose vs. task-specific
>
> We respectfully disagree that our paper misleads readers, as the task-specific scope has been explicitly mentioned in multiple sections of the paper: **L96 (introduction), L295 (experiments), and L508 (related work)**. That said, we acknowledge the value of further emphasizing this scope and its limitations and will revise the manuscript to ensure they are more clearly articulated.
>
>
> We’d also like to clarify that our goal is not to claim performance superiority over RLAIF but to highlight the trade-offs between performance and computational efficiency. RLAIF is powerful but requires (1) heavy prompt engineering and (2) expensive compute. **Our framework provides practitioners with an additional choice, and can co-exist with RLAIF in complementary way**: for task-specific datasets with limited human feedback, our method offers a cost-effective, scalable alternative to GPT-4.
>
> The task-specific nature of our method does not diminish its value; instead, it underscores its practical utility in addressing real-world constraints. We believe our work remains compelling for practitioners handling large-scale data who cannot justify the expense of resource-intensive models or may favor light models for sustainability concerns.
>
> ------------
> > While I agree that using a weaker LLM has potential efficiency advantages, I remain skeptical of its claimed superiority in effectiveness.
>
> We would like to reiterate that **our paper is not about claiming superiority**. Our central claim "_The alignment performance using weak LLM feedback **closely matches** or even surpasses that of using human feedback._" is supported by both qualitative and quantitative evidence, including additional experiments:
>
> -  Feedback from GPT-4 with more advanced prompt or high-quality human labels can indeed enhance the student model's performance, which is an expected conclusion. However, it is important to note that both comes with cost. **Collecting GPT-4 feedback with higher-quality prompts often demands greater computational resources and relies heavily on prompt engineering. Similarly, obtaining high-quality human labels is more labor-intensive.** In practical applications, balancing efficiency and effectiveness could be a critical consideration. Notably, our findings reveal that with weak LLM feedback achieves significantly higher efficiency while delivering comparable performance to the student model learned from the feedback from GPT-4 with more advanced prompt or high-quality human labels.
> - Using higher-quality human feedback data, using weak LLM feedback achieves gold reward 5.23, _which closely matches that of using human feedback 5.19_. This is consistent with our central claim.
> - Our human evaluation indicates a win-rate of 52%, closely aligning with our main findings that weak LLM feedback _is at least as effective as human feedback_. The same trend can be observed using gold reward model based evaluations. Across all five models, weak LLM feedback _consistently demonstrates close or slightly better performance than human feedback_.
> - Section 3.3 demonstrates that alignment performance is **nearly comparable** across different supervisor sizes, suggesting that supervisor size has a less significant impact on feedback effectiveness. We additionally provide experiments using Llama-2-13B (due to computational constraints, we cannot train larger than 13B) as the supervisor for comparison. Additional experiments further confirm this: using Llama-2-13B achieves nearly identical performance to OPT-125M (7.90 vs. 7.93).
>
>
> | Feedback              | Gold reward |
> |---------------------------|-------|
> |OPT-125M     | 7.90  |
> |OPT-1.3B    | 7.82  |
> |OPT-2.7B    | 7.96  |
> |Llama-2-13B     | 7.93  |
> | Human Labeler  | 7.19  |

---

> > ### Comment · Reviewer_1dzH · 2024-11-25
> > **Response to Authors**
> >
> > Thanks for the detailed explanation and additional experiments. Hope the clarification for the superiority in specific domains and more solid experimental results can be included in the revision. I have increased my rating accordingly. Thank you!

---

> > > ### Author Response · Authors · 2024-11-25
> > > **Thanks for the follow up**
> > >
> > > Dear Reviewer 1dzH
> > >
> > > Thank you for your feedback and for increasing your rating. We will ensure the revision according all the reviewers' suggestions. Your comments has been invaluable in improving our work.
> > >
> > > Best,
> > > Authors.

---

### Author Response · Authors · 2024-11-18
**General Response**

We sincerely thank all the reviewers for their time and insightful feedback. We are encouraged by the enthusiastic reception of our work, with reviewers recognizing it as both **critical and timely** in advancing the field of LLM alignment (1dzH, oS4F, 2XYr). Reviewers recognize our framework to be **cost-effective, promising, scalable, and has significant impact and implications on LLM alignment practices** (1dzH, oS4F, ntey, ZoiK), supported by **comprehensive** and **convincing experiments** (oS4F, ntey, ZoiK, 2XYr). Furthermore, reviewers commended our paper to be **clear, polished, and well-written** (1dzH, ntey, ZoiK).

The importance of our work is underscored by its potential to address fundamental trade-offs between human annotation costs and computational expense in alignment feedback. Below, we summarize the key contributions of our work:

- Our study addresses a significant challenge in LLM alignment—reducing the reliance on costly human and high-capacity model annotations. _Our work is the first to demonstrate that weak LLMs can rival and even surpass humans in alignment feedback_.
- Our work is supported by extensive experiments across various model families and scales, showcasing the generalizability and effectiveness of the proposed framework. The results highlight the practical viability of using weak LLMs to achieve high alignment quality without requiring the latest or most powerful models (ntey, ZoiK, 2XYr).
- Our findings, along with an in-depth understanding of weak LLM feedback, suggest that alignment can be achieved efficiently and at a lower cost, making this approach more scalable and accessible for real-world applications. The proposed method has the potential to shape future LLM alignment practices significantly (oS4F, ntey, ZoiK).

We address each reviewer's comments in detail below, incorporating additional experiments and analyses as suggested. These new results further reinforce our findings. We are committed to revising the manuscript according to the reviewers' suggestions, and we believe these improvements will strengthen the impact of our work.

---

### Meta-Review · Area_Chair_LpJb · 2024-12-20

**Metareview:**

This work focuses on LLM alignment and proposes a method that employs a semi-supervised framework. In this framework, a weak LLM, trained on a small labeled dataset, provides preference feedback for a larger unlabeled dataset. Experiments show its effectiveness in alignment tasks and validate the quality of feedback provided by weak LLMs.

**Additional Comments On Reviewer Discussion:**

Discussion summary during the rebuttal period:
1. **Novelty Compared to RLAIF**: The authors differentiate their method by using a weak LLM that is tuned on a human-labeled alignment dataset to produce AI feedback. This is not entirely novel but the paper is still commendable overall.

2. **Need for More Comprehensive Experiments**: The authors add more detailed experiments, such as using RewardBench to assess the quality of feedback, and conducting evaluations on more complex tasks along with comparisons using more advanced prompts.

Recommendation: The revised version should clearly outline the main argument and include additional experiments during the rebuttal period to enhance the persuasiveness and provide more insights.

---

### Decision · Program_Chairs · 2025-01-22

Accept (Poster)